# How about Levetiracetam in Glioblastoma? An Institutional Experience and Meta-Analysis

**DOI:** 10.3390/cancers13153770

**Published:** 2021-07-27

**Authors:** Ramazan Jabbarli, Yahya Ahmadipour, Laurèl Rauschenbach, Alejandro N. Santos, Marvin Darkwah Oppong, Daniela Pierscianek, Carlos M. Quesada, Sied Kebir, Philipp Dammann, Nika Guberina, Björn Scheffler, Klaus Kaier, Martin Stuschke, Ulrich Sure, Karsten H. Wrede

**Affiliations:** 1Department of Neurosurgery and Spine Surgery, University Hospital Essen, 45147 Essen, Germany; yahya.ahmadipour@uk-essen.de (Y.A.); laurel.rauschenbach@uk-essen.de (L.R.); alejandro.santos@uk-essen.de (A.N.S.); Marvin.DarkwahOppong@uk-essen.de (M.D.O.); daniela.pierscianek@uk-essen.de (D.P.); philipp.dammann@uk-essen.de (P.D.); ulrich.sure@uk-essen.de (U.S.); karsten.wrede@uk-essen.de (K.H.W.); 2German Cancer Consortium (DKTK) Partner Site, University Hospital Essen, 45147 Essen, Germany; sied.kebir@uk-essen.de (S.K.); nika.guberina@uk-essen.de (N.G.); bjoern.scheffler@uk-essen.de (B.S.); martin.stuschke@uk-essen.de (M.S.); 3DKFZ-Division Translational Neurooncology at the WTZ, German Cancer Research Center (DKFZ) Heidelberg & German Cancer, Consortium (DKTK) Partner Site, University Hospital Essen, 45147 Essen, Germany; 4Department for Neurology, University Hospital Essen, 45147 Essen, Germany; carlos.quesada@uk-essen.de; 5Department of Neurology, Division of Clinical Neurooncology, University Hospital Essen, 45147 Essen, Germany; 6Department of Radiotherapy, University Hospital Essen, 45147 Essen, Germany; 7Institute of Medical Biometry and Statistics, Faculty of Medicine and Medical Center, University of Freiburg, 79106 Freiburg, Germany; klaus.kaier@uniklinik-freiburg.de

**Keywords:** glioblastoma, survival, epilepsy, levetiracetam, tumor, progression

## Abstract

**Simple Summary:**

To date, there is a discrepancy regarding the role of antiepileptic drugs on glioblastoma survival. In the present study, based on large institutional cohort and enhanced with a meta-analysis of seven previously published studies, we show a robust association between the perioperative start of levetiracetam treatment with increased overall and progression-free survival in glioblastoma. Our results encourage the initiation of a prospective clinical trial to analyze the antitumor effect of levetiracetam in glioblastoma patients.

**Abstract:**

Despite multimodal treatment, the prognosis of patients with glioblastoma (GBM) remains poor. Previous studies showed conflicting results on the effect of antiepileptic drugs (AED) on GBM survival. We investigated the associations of different AED with overall survival (OS) and progression-free survival (PFS) in a large institutional GBM cohort (*n* = 872) treated January 2006 and December 2018. In addition, we performed a meta-analysis of previously published studies, including this study, to summarize the evidence on the value of AED for GBM prognosis. Of all perioperatively administered AED, only the use of levetiracetam (LEV) was associated with longer OS (median: 12.8 vs. 8.77 months, *p* < 0.0001) and PFS (7 vs. 4.5 months, *p* = 0.001). In the multivariable analysis, LEV was independently associated with longer OS (aHR = 0.74, *p* = 0.017) and PFS (aHR = 0.68, *p* = 0.008). In the meta-analysis with 5614 patients from the present and seven previously published studies, outcome benefit for OS (HR = 0.83, *p* = 0.02) and PFS (HR = 0.77, *p* = 0.02) in GBM individuals with LEV was confirmed. Perioperative treatment with LEV might improve the prognosis of GBM patients. We recommend a prospective randomized controlled trial addressing the efficacy of LEV in GBM treatment.

## 1. Introduction

Glioblastoma (GBM) is the most aggressive and frequent primary brain tumor [1]. The standard of care for GBM patients includes microsurgical tumor resection followed by concomitant chemoradiotherapy (CCRT) with temozolomide (TMZ) followed by adjuvant TMZ therapy [2,3]. Despite multimodal treatment, median survival after GBM diagnosis is limited to 14–16 months, with the survival following progression at only 6–8 months [4].

Several survival markers for GBM have been identified so far, such as the patients’ age, initial clinical condition, extent of resection (EOR) and, particularly, molecular tumor characteristics such as methylation of the O6-methylguanin-DNA-methyltransferase (MGMT) gene promotor, or mutation of the isocitrate-dehydrogenase 1 (IDH1) gene [5]. Early seizures and treatment with antiepileptic drugs (AED) are also associated with GBM survival [6,7,8,9,10,11,12]. However, it remains unclear whether the supposed survival benefit is related to earlier diagnosis and treatment or (direct or indirect) the intrinsic antitumor activity of AED [13]. Moreover, several recent studies could not confirm improved survival in GBM patients with early epilepsy/AED treatment [14,15,16,17,18,19,20,21,22,23].

Due to a large number of routinely used AED, the reported discrepancies in outcome effect might be at least partially related to individual AED pharmacokinetics. In particular, enzymes inducing AED (EIAED) like carbamazepine, phenytoin, and phenobarbital were reported to alter the effect of some antitumor agents [12,24]. At the same time, the most common chemotherapeutic agent in GBM, TMZ, is not significantly metabolized by the CYP450 hepatic system, thus limiting the possibility of interactions with EIAED [1]. Previous reports on the survival effect of nonenzyme-inducing AED (NEIAED), such as valproic acid (VPA) and levetiracetam (LEV), have also shown inconsistent results [9,14,25,26,27], not allowing definite recommendations. Therefore, the real impact of AED on the prognosis of GBM requires further clarification.

Using a large institutional observational cohort, we investigated the associations of different AED with overall survival (OS) and progression-free survival (PFS) of GBM. In addition, we performed a meta-analysis of previously published studies, including this study, to summarize the evidence on the value of AED for GBM prognosis.

## 2. Materials and Methods

### 2.1. Institutional Cohort

#### 2.1.1. Patient Population

This retrospective study was based on an institutional observational GBM database and performed according to the STROBE guidelines. The Institutional Ethics Committee approved the study. All consecutive cases with newly diagnosed GBM treated at our institution between January 2006 and December 2018 were eligible for the study. The exclusion criteria were: (1) pediatric cases (<18 years old, *n* = 7); (2) extracranial location (*n* = 1).

#### 2.1.2. GBM Management

Histological evaluation following the 2016 Classification of the Central Nervous System Tumors of the World Health Organization confirmed the diagnosis after stereotactic biopsy or tumor resection [28]. Early postoperative magnetic resonance imaging (MRI) within 72 h after tumor resection was performed to assess the EOR. The absence of an enhancing lesion on T1-weighted contrast-enhanced images was defined as a gross-total resection, and the remaining cases were regarded as debulking. 

Standard postoperative treatment included CCRT with TMZ and adjuvant TMZ [3]. Patients underwent repeated follow-ups with MRI and, if necessary, positron emission tomography (PET) imaging at different time intervals (every 2–3 months, or earlier upon clinical deterioration). The occurrence of tumor progression was assessed according to the recent Response Assessment in Neuro-Oncology (RANO) criteria [29].

#### 2.1.3. Epilepsy Treatment

Perioperative management of GBM-associated epilepsy and indication for AED treatment are described elsewhere [13]. In short, AED treatment was usually indicated only for symptomatic cases. In individuals with prophylactic AED initiated in the referring hospital, and in patients with previous AED treatment due to known epilepsy, the same AED medication was continued. The choice of AED was based on the preference of consultant epileptologists and the neurologists from the referring hospitals.

#### 2.1.4. Data Management

The cases with AED medication initiated prior to the start of postoperative adjuvant treatment were recorded, including the indication (symptomatic vs. prophylactic AED treatment) and the generic name. AED treatment initiated due to epilepsy occurring after the start of CCRT was not considered during the present analysis. The following outcome-relevant parameters were collected for further analysis: age, Karnofsky Performance Scale (KPS) score at admission, IDH1-mutation and MGMT-promoter methylation status, EOR (biopsy vs. tumor debulking vs. gross-total resection), and postoperative treatment. Finally, the parameters of OS and PFS were recorded from the follow-up data.

#### 2.1.5. Study Endpoints and Statistical Analysis

The effect of different AED on OS and PFS were the study endpoints. Depending on the used AED, their frequency in the cohort and resulting significances, the survival data were analyzed in different AED-related categories: (1) AED vs. No AED; (2) mono vs. combined-AED; (3) EIAED vs. NEIAED; (4) LEV vs. VPA vs. any other AED (Not LEV/VPA) vs. No AED; (5) LEV vs. any other AED (Not LEV); (6) LEV vs. No LEV (other AED + No AED). Survival differences between the AED categories were analyzed using Kaplan-Meier survival plots with long-rank tests and Cox regression analysis. The analyses were performed in the whole cohort and in the subgroup with standard postoperative treatment (CCRT + TMZ-cohort). Finally, the association between significant AED and GBM survival data was assessed using multivariable Cox regression analysis adjusted for patients’ age, indication to AED treatment, KPS score, tumor location, EOR, MGMT methylation and IDH1 mutation status, and postoperative adjuvant treatment. Statistical analyses were performed with the PRISM (version 5.0, GraphPad Software Inc., San Diego, CA, USA) and SPSS (version 25, SPSS Inc., IBM, Chicago, IL, USA) software packages. The missing data in the database were replaced using multiple imputation. Differences with a *p* < 0.05 were regarded as statistically significant. Confidence intervals (95% CI) were not adjusted for multiple comparisons, and inferences drawn from them may not be reproducible. Survival data were reported in median values, including 95% CI. 

### 2.2. Literature Review and Meta-Analysis

To summarize the evidence on survival impact of the most significant study results, we systematically searched PubMed, PubMed Central, Scopus, EMBASE, Web of Science and Cochrane Library databases. We identified all studies published before 1 March 2021 that reported on the associations between the AED of interest and OS/PFS in GBM patients. To assess eligibility of the studies, RJ and YA independently screened the titles and abstracts and, if necessary, the full text and the reference list of relevant publications for additional articles. The detailed search strategy and results are presented in Appendix A with the search terms, and Appendix A with the flow-chart. The review was restricted to English-language studies. The extracted data included publication year, geographic origin, number of patients in each arm and appropriate OS/PFS data as reported by the authors (median values, 95% CI, and/or (adjusted) hazard ratios [(a)HR]). 

A formal meta-analysis was performed using Review Manager (version 5.4.1, Nordic Cochrane Centre, Copenhagen, Denmark). Because of assumed heterogeneity, we used random-effects models of meta-analysis (Mantel-Haenzsel method). PRISMA recommendations were followed for this meta-analysis.

## 3. Results

### 3.1. Patient Population

The institutional cohort included 872 cases with newly diagnosed GBM. The median age of the cohort was 65.3 years (range: 19.8–91.5 years) and 507 individuals were males (58.1%). CCRT with TMZ was initiated in 646 patients (74.1%, CCRT + TMZ-cohort). Median OS was 9.7 months (95% CI: 8.81–10.6) and 12.4 months (95% CI: 11.5–13.31 months) in the whole cohort and CCRT + TMZ subgroup, respectively. Accordingly, the median PFS was 5 (95% CI: 4.53–5.47) and 6 months (95% CI: 5.49–6.51), respectively. Detailed information on baseline population characteristics is shown in Appendix A. Data on the cases with AED treatment (*n* = 295, 33.8%) including the indications to and the list of used AEDs, are presented in Appendix A. In short, eleven different AED were used in the cohort; NEIAED (n = 257, 87%). LEV (n = 196, 65%) and VPA (n = 48, 16%) were the most frequently prescribed AED. Usually, monotherapy was sufficient for perioperative seizure control, whereas only 11 patients required combined AED treatment. 

### 3.2. AED and OS

Patients with AED revealed better median OS (11.47 months, 95% CI: 9.63–13.3) than GBM patients without AED treatment (8.73 months, 95% CI: 7.66–9.81, *p* = 0.001). Of all separately assessed AED, only LEV treatment was associated with favorable OS (12.8 months, 95% CI: 10.82–14.78) as compared to GBM individuals with any other AED (9.07 months, 95% CI: 6.68–11.46, *p* = 0.004) or no LEV (8.77 months, 95% CI: 7.77–9.77, *p* < 0.0001, see also Figure 1 with the Kaplan-Meier-survival plots and Table 1 with OS data to most relevant AED categories).

In the CCRT + TMZ-subgroup, LEV treatment was significantly associated with OS: 15 months (95% CI: 12.34–17.68) vs. 12.13 months (95% CI: 11.24–13.03) in individuals without LEV (*p* = 0.002). Finally, multivariable Cox regression analysis confirmed an independent association between LEV treatment and OS of GBM patients in the institutional cohort (aHR = 0.77, 95% CI: 0.61–0.98, *p* = 0.037, see Table 2). An additional subcohort analysis restricted to the IDH-wild-type GBM cases also showed an independent impact of LEV treatment on OS (aHR = 0.64, 95% CI: 0.43–0.96, *p* = 0.032, see Appendix A).

### 3.3. AED and PFS

There was a significant difference in PFS between patients with (6 months, 95% CI: 5.05–6.95) and without (4 months, 95% CI: 3.27–4.73, *p* = 0.009) AED treatment. The comparison of PFS in different AED categories (Table 3) showed PFS benefit with NEIAED (6 months, 95% CI: 5–7) over EIAED (4 months, 95% CI: 1.43–6.57, *p* = 0.025), as well as better PFS with LEV (7 months, 95% CI: 5.83–8.17) vs. any other AED (4 months, 95% CI: 2.79–5.21, *p* = 0.007) or no LEV (4.5 months, 95% CI: 3.86–5.14, *p* = 0.001, see also Figure 2 with the Kaplan-Meier survival plots).

In the CCRT + TMZ-subgroup, PFS difference remained significant only for LEV (8 months, 95% CI: 6.24–9.76) vs. no LEV (6 months, 95% CI: 5.46–6.54, *p* = 0.024). The multivariable Cox regression analysis also showed an independent association between LEV treatment and PFS (aHR = 0.71, 95% CI: 0.53–0.95, *p* = 0.022, Table 2)

### 3.4. Meta-Analysis

The data from seven previous publications (including one study with pooled analysis) were extracted [14,25,26,27,30,31,32], and together with the findings of the present study, were included in the meta-analysis with a total of 5614 GBM patients. 

The results of the meta-analysis confirmed better OS (HR = 0.83, 95% CI: 0.71–0.97, *p* = 0.02, Figure 3) and PFS (HR = 0.77, 95% CI = 0.62–0.96, *p* = 0.02, Figure 4) with LEV. An additional meta-analysis based on five studies reporting the results of multivariable analysis (aHR = 0.68, 95% CI = 0.50–0.94, *p* = 0.02, see Appendix A) and survival differences (+3.87 months, 95% CI = 1.12–6.61 months, *p* = 0.006, see Appendix A), also confirmed the survival benefit in GBM individuals with LEV.

## 4. Discussion

AED treatment was previously reported to be associated with better survival of GBM patients; however, this topic remains controversial. In our large institutional GBM series, we analyzed the impact of various AED on the prognosis of GBM. Only treatment with LEV showed strong associations with OS and PFS. The additional meta-analysis confirmed the survival benefit of LEV for GBM.

### 4.1. AED and GBM Prognosis: Direct or Indirect Antitumor Effect, Coincidence or Myth?

A large number of studies have addressed possible interactions between early epileptic seizures and AED treatment with GBM survival [2,6,7,8,9,10,11,14,15,16,17,18,19,20,21,22,23,25,30,32,33,34,35]. Due to conflicting study results, there is no consensus about the validity of the link between early seizures and GBM prognosis, let alone the causal background of this relationship. Considerable heterogeneity of the published data concerning potential effector (seizures vs. AED/early vs. any seizures/nonunique indications/various AED) and analyzed cohorts (different study designs and GBM sub-populations) limits the possibility of cumulative conclusions based on the available literature.

As to the pathophysiologic mechanisms which might explain the impact of GBM-associated epilepsy on patients’ survival, there are two basic hypotheses. First, seizures at onset might lead to earlier diagnosis and rapid treatment of GBM [13]. On the other side, the direct or indirect (via interaction with chemotherapy) antitumor effect of AED has also been widely discussed in the literature [36]. 

In this respect, VPA is probably the most intensely analyzed AED in GBM cohorts. The possible antitumor effect of VPA might be related to its radiotherapy-sensitizing properties due to the inhibition of histone deacetylase enzyme, enhancement of cellular redox reactions (in combination with chemotherapy) and reduced TMZ clearance [1,2,12]. However, a recent pooled analysis [14] failed to show survival benefit from VPA in GBM patients. In addition, VPA use might be associated with additional harm related to the risk of thrombocytopenia and neutropenia or platelet dysfunction [1].

### 4.2. LEV: A Light at the End of the Tunnel?

LEV is another promising AED that is well known, and not only for its remarkable antiepileptic features. The potential antitumor activity of LEV might be conditioned through epigenetic silencing of the enzyme MGMT, and subsequent increase of the efficacy of TMZ [12]. However, despite several positive reports from single-center retrospective series [26,27,30,31,32], Happold et al. [14] could not confirm a better outcome with LEV in their pooled analysis. 

Of all tested AED, only LEV revealed a robust association with better OS and PFS of GBM patients in the institutional cohort, regardless of the outcome-relevant baseline characteristics (such as age, KPS score, tumor location, EOR and molecular tumor characteristics), indications to AED (prophylactic or due to early seizures) and postoperative treatment. An additional meta-analysis with 5416 patients from eight studies substantiated the survival benefit of LEV for GBM. 

As to other widely accepted outcome predictors of GBM, our study confirmed independent associations between patients’ age, KPS, EOR, MGMT-methylation status and adjuvant treatment with postoperative survival. At the same time, IDH1 failed to show significant results in the multivariable analysis. This might be related to the fact that only 3% of the analyzed GBM cohort presented with an IDH mutation. Therefore, in a multivariable regression analysis containing nine parameters, IDH mutation status could not reach the significance level. 

Although this study provides evidence for LEV treatment in GBM patients, some limitations must be considered. Due to the retrospective design of our study, some selection and information bias cannot be entirely ruled out. Similar to observations from previous studies, there was a heterogenous prescription pattern of AED in the analyzed institutional cohort. Of eleven different AED, LEV and VPA were the most commonly selected drugs, accounting together for >80% of AED prescriptions. Although there were no patients with an AED switch, and the combined use of AED was required in only 1% of cases, such over and under representation patterns strongly limit the comparability of the used AEDs with regard to the impact on postoperative survival. We could not address the prognostic value delayed AED treatment due to secondary epilepsy given after the initiation of CCRT, since these data were missing for the majority of the patients. On the other side, the later beginning of antiepileptic treatment might underpower the potency of the antitumor effect of AED, and particularly its sensitizing impact on chemoradiation. Therefore, only the cases with early initiation of AED were able to reflect the role of AED on GBM outcome.

As to the meta-analysis, the included studies exhibited some structural and methodological heterogeneity, limiting the value of cumulative conclusions from this pooled data. Despite these limitations, our study presents strong evidence encouraging the initiation of a prospective clinical trial to analyze the antitumor effect of LEV in GBM patients.

## 5. Conclusions

Of all addressed AED, only LEV showed significant associations with OS and PFS, regardless of the patients’ tumor characteristics and postoperative treatment. The additional meta-analysis confirmed the survival benefit of LEV for GBM patients. We recommend a prospective randomized controlled trial addressing the efficacy of LEV in GBM treatment.

## Figures and Tables

**Figure 1 cancers-13-03770-f001:**
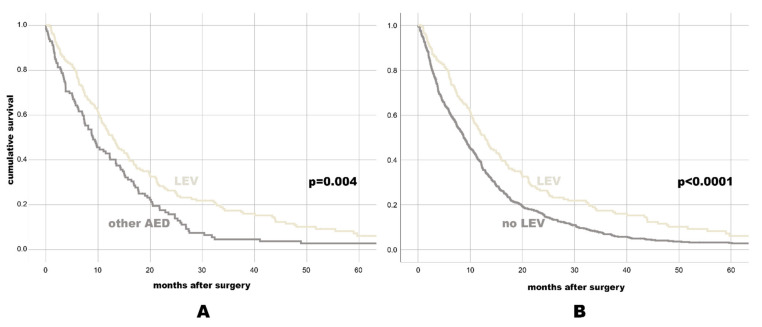
Kaplan-Meier-survival plot for OS for individuals with LEV vs. any other AED (**A**), and for individuals with LEV vs. No LEV (**B**).

**Figure 2 cancers-13-03770-f002:**
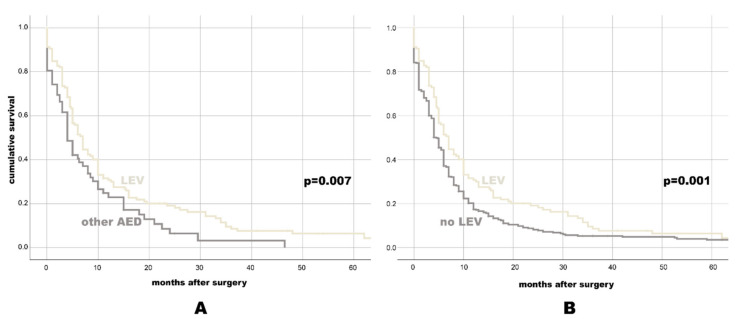
Kaplan-Meier-survival plot for PFS for individuals with LEV vs. any other AED (**A**), and for individuals with LEV vs. No LEV (**B**).

**Figure 3 cancers-13-03770-f003:**
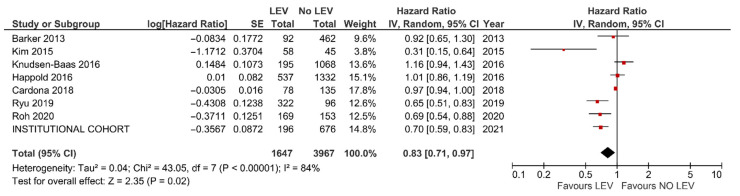
Meta-analysis of studies reporting on OS of GBM patients with and without LEV treatment.

**Figure 4 cancers-13-03770-f004:**
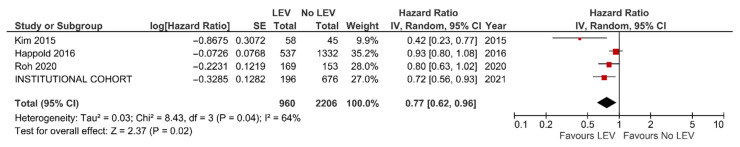
Meta-analysis of studies reporting on PFS of GBM patients with and without LEV treatment.

**Table 1 cancers-13-03770-t001:** OS values incl. HRs in different AED categories within the whole cohort and subgroup with adjuvant chemoradiation with TMZ (CCRT + TMZ-subgroup).

	In the Whole Cohort					In the CCRT + TMZ-Subgroup				
	Median	95% CI	HR	95% CI	*p*-Value	Median	95% CI	HR	95% CI	*p*-Value
Mono AED vs. Combined AED										
Mono AED	15.0	13.01–16.99	1.23	0.65–2.33	0.532	15.0	13.01–16.99	1.23	0.65–2.33	0.532
Combined AED	10.27	7.94–12.59				10.27	7.94–12.59			
NEIAED vs. EIAED										
NEIAED	11.8	9.96–13.64	0.89	0.61–1.28	0.520	14.5	12.45–16.55	1.17	0.71–1.92	0.532
EIAED	9.67	4.67–14.67				15.07	11.42–18.72			
LEV vs. Any Other AED										
LEV	12.8	10.82–14.78	0.70	0.54–0.89	0.004	15	12.32–17.68	0.78	0.59–1.05	0.100
Other AED	9.07	6.68–11.46				14.23	10.27–18.19			
LEV vs. No LEV										
LEV	12.8	10.82–14.78	0.68	0.57–0.81	<0.0001	15	12.32–17.68	0.73	0.60–0.89	0.002
No LEV	8.77	7.77–9.77				12.13	11.24–13.03			

Abbreviations: CI—confidence interval. HR—hazard ratio.

**Table 2 cancers-13-03770-t002:** Multivariable Cox regression analysis for the association between LEV medication and OS/PFS of GBM in the whole cohort.

Parameter	OS			PFS		
	aHR	95% CI	*p*-Value	aHR	95% CI	*p*-Value
LEV treatment	0.76	0.59–0.97	0.030	0.69	0.51–0.93	0.015
Age. per-year-increase	1.03	1.02–1.03	<0.0001	1.02	1.01–1.03	<0.0001
AED due to seizures	1.10	0.89–1.36	0.381	1.21	0.93–1.58	0.161
KPS < 80%	1.40	1.18–1.65	<0.0001	1.32	1.08–1.62	0.006
Tumor location (midline)	1.33	0.93–1.90	0.111	1.30	0.86–1.99	0.211
EOR	0.63	0.57–0.70	<0.0001	0.73	0.64–0.82	<0.0001
MGMT-methylation	0.68	0.57–0.81	<0.0001	0.77	0.63–0.93	0.007
IDH1-mutation	0.76	0.38–1.54	0.388	0.99	0.43–2.30	0.984
CCRT + TMZ	0.38	0.31–0.47	<0.0001	0.32	0.26–0.41	<0.0001

Abbreviations: OS—overall survival, PFS—progression-free survival, aHR—adjusted hazard ratio, CI—confidence interval, LEV—levetiracetam, KPS—Karnofsky Performance Scale score, EOR—extent of resection (biopsy vs. debulking vs. gross-total resection), MGMT—O[6]-methylguanine-DNA methyltransferase promoter methylation, IDH1—Isocitrate dehydrogenase 1 mutation, CCRT—concomitant chemoradiotherapy, TMZ—temozolomide.

**Table 3 cancers-13-03770-t003:** PFS values incl. HRs in different AED categories within the whole cohort and subgroup with adjuvant chemoradiation with TMZ (CCRT + TMZ-subgroup).

	In the Whole Cohort					In the CCRT + TMZ-Subgroup				
	Median	95% CI	HR	95% CI	*p*-Value	Median	95% CI	HR	95% CI	*p*-Value
Mono AED vs. Combined AED										
Mono AED	8.0	6.4–9.6	1.33	0.62–2.86	0.464	8.0	6.4–9.6	1.33	0.62–2.86	0.464
Combined AED	4.0	1.52–6.48				4.0	1.52–6.48			
NEIAED vs. EIAED										
NEIAED	6.0	5.0–7.0	0.59	0.37–0.96	0.034	8.0	6.37–9.63	0.83	0.41–1.71	0.620
EIAED	4.0	1.43–6.57				5.0	3.54–6.46			
LEV vs. Any Other AED										
LEV	7.0	5.83–8.17	0.67	0.50–0.91	0.010	8.0	6.24–9.76	0.80	0.55–1.17	0.249
Other AED	4.0	2.79–5.21				7.0	3.32–10.68			
LEV vs. No LEV										
LEV	7.0	5.83–8.17	0.72	0.59–0.88	0.001	8.0	6.24–9.76	0.77	0.62–0.97	0.024
No LEV	4.5	3.86–5.14				6.0	5.46–6.54			

Abbreviations: CI—confidence interval. HR—hazard ratio.

## Data Availability

Any data not published within the article will be shared in anonymized manner by request from any qualified investigator.

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
