# Peer review of "How about Levetiracetam in Glioblastoma? An Institutional Experience and Meta-Analysis"

_cancers, 2021, doi:10.3390/cancers13153770_

Round 1

Reviewer 1 Report

  1. Line 131-133: I wonder why you missed EMBASE when you performed the systematic review. I think it is very important database. Also, search strategy.
  2. Table S2: Please explain which kinds of antiepileptic drugs (AED) are used. I think many kinds of antiepileptic drugs are tried and I wonder how many patients received more than 1 AED concomitantly and how often AEDs are switched. I mean, please explain overall status of AEDs based on your analysis. (1) AED vs No AED; 2) mono- vs combined-114 AED; 3) EIAED vs NEIAED; 4) LEV vs VPA vs any other AED (Not LEV/VPA) vs No 115 AED; 5) LEV vs any other AED (Not LEV); 6) LEV vs No LEV (other AED + No AED).
  3. Discussion: please add a comment on the prescription pattern in this study in discussion.
  4. Discussion: please add a comment on the result of MGMT methylation and IDH1-mutation.

Author Response

  1. Line 131-133: I wonder why you missed EMBASE when you performed the systematic review. I think it is very important database. Also, search strategy.

Response: We thank the reviewer for this important remark. Indeed, EMBASE presents one of the most powerful academic databases. As recommended by the reviewer, we included EMBASE in the literature review. In EMBASE, 43 studies fulfilled the search criteria. However, there was no additional study eligible for inclusion in the meta-analysis. The inclusion of a new academic database in the literature review was reflected in the revised manuscript (updated Figure S1 and line 132 in the manuscript text). The search strategy was described in the supplementary Table S1.

  1. Table S2: Please explain which kinds of antiepileptic drugs (AED) are used. I think many kinds of antiepileptic drugs are tried and I wonder how many patients received more than 1 AED concomitantly and how often AEDs are switched. I mean, please explain overall status of AEDs based on your analysis. (1) AED vs No AED; 2) mono- vs combined-114 AED; 3) EIAED vs NEIAED; 4) LEV vs VPA vs any other AED (Not LEV/VPA) vs No 115 AED; 5) LEV vs any other AED (Not LEV); 6) LEV vs No LEV (other AED + No AED).

Response: We agree with the reviewer that the detailed information on the AEDs used in the analyzed cohort is of eminent importance for understanding the study results. The supplementary Figure S2 summarizes the most important information on the indications to and the list of used AEDs in the cohort, including the prescription pattern and the need for combined antiepileptic treatment. In particular, only 1.4% of the whole cohort required combined AED treatment. Of them, no AED switch was practiced.

  1. Discussion: please add a comment on the prescription pattern in this study in discussion.

Response: As recommended by the reviewer, we included additional information regarding the prescription pattern in the manuscript body (in the results and in the discussion part of the manuscript).

  1. Discussion: please add a comment on the result of MGMT methylation and IDH1-mutation.

Response: As recommended by the reviewer, the results on the predictive value of molecular-genetic markers were discussed in the revised version of the manuscript.

Reviewer 2 Report

Jabbarli et al. report a large institutional cohort with a meta-analysis of the literature on the impact of levetiracetam on overall survival and progression-free survival in glioblastoma. The manuscript is overall clear and well-written. I have few minor comments.

  • page 2, line 84-86: it is not clearly stated wether cases diagnosed between 2006 and 2016 were histologically revised according to the 2016 WHO classification. In this case, it is probably more appropriate to distinguish in the following analyses between IDH-wildtype and IDH-mutant GBM, since they are different entities;
  •  Table S2, which reports baseline features of the patients included in the cohort, should also include tumor location (es. supratentorial hemispheric vs infratentorial vs diencephalic?) and whether bilateral and/or multifocal. The variable "location" should also be included in multivariate analyses (infratentorial or midline GBM are unlikely to need AEDs for seizures and they do have poorer prognosis). I would also suggest to add to table S2 the different AEDs used in the institutional cohort, to have a grasp of the "competitors" of levetiracetam and of the size of each subgroup. This is important to be aware of the robustness of statistical analyses
  • I found table S3 and S4 very informative and I think the manuscript would benefit from displaying them as ordinary material. Perhaps figure 1 and figure 2 could be merged (as well as Figure 3 and Figure 4) to make room to include them.

Author Response

  1. page 2, line 84-86: it is not clearly stated wether cases diagnosed between 2006 and 2016 were histologically revised according to the 2016 WHO classification. In this case, it is probably more appropriate to distinguish in the following analyses between IDH-wildtype and IDH-mutant GBM, since they are different entities;

Response: We are grateful to the reviewer for this essential comment. Indeed, the (immune-)histological and molecular genetic data on all cases in the analyzed cohort were reviewed in accordance with the 2016 WHO classification. As suggested by the reviewer, we performed an additional sub-cohort analysis by separating the cohort according to the IDH status. Therefore, a significant association between LEV treatment and postoperative survival could be confirmed in the additional multivariable analysis restricted to the IDH-wild-type glioblastoma cases. Due to low prevalence of IDH-mutant cases in the cohort (3%), we did not perform another multivariable analysis in this very small subgroup of glioblastoma cases.

  1.  Table S2, which reports baseline features of the patients included in the cohort, should also include tumor location (es. supratentorial hemispheric vs infratentorial vs diencephalic?) and whether bilateral and/or multifocal. The variable "location" should also be included in multivariate analyses (infratentorial or midline GBM are unlikely to need AEDs for seizures and they do have poorer prognosis). I would also suggest to add to table S2 the different AEDs used in the institutional cohort, to have a grasp of the "competitors" of levetiracetam and of the size of each subgroup. This is important to be aware of the robustness of statistical analyses

Response: We agree with the reviewer on the relevance of tumor location for study results. Therefore, we included the information on the tumor location in the supplementary Table S2. Moreover, we included tumor location as an additional covariant in our multivariable Cox regression analyses. As to the detailed information on the used AEDs, the appropriate information is presented in the supplementary Figure S2.

  1. I found table S3 and S4 very informative and I think the manuscript would benefit from displaying them as ordinary material. Perhaps figure 1 and figure 2 could be merged (as well as Figure 3 and Figure 4) to make room to include them.

Response: As recommended by the reviewer, we moved the supplementary Tables S3 and S4 to the main manuscript (as Tables 1 and 2) and accordingly renumbered the former table 1 (now Table 3).